# Usability of augmented reality technology in tele-mentorship for managing clinical scenarios—A study protocol

**Dung T. Bui** [ID]¹*, **Tony Barnett**¹, **Ha Hoang**¹, **Winyu Chinthammit**²

**1** Centre for Rural Health, School of Health Sciences, College of Health and Medicine, University of Tasmania, Launceston, Tasmania, Australia, **2** Human Interface Technology Laboratory, School of Information and Communications Technology, University of Tasmania, Launceston, Tasmania, Australia

* dungtrung.bui@utas.edu.au

**Funding:** The authors acknowledge the support received from the University of Tasmania and the Commonwealth Government Department of Health Rural Health Multidisciplinary Training program.

## Abstract

### Background

Tele-mentorship is considered to offer a solution to training and providing professional assistance at a distance. Tele-mentoring is a method in which a mentor interactively guides a mentee at a different geographic location in real time using a technological communication device. During a healthcare procedure, tele-mentoring can support a medical expert, remote from the treatment site, to guide a less-experienced practitioner at a different geographic location. Augmented Reality (AR) technology has been incorporated in tele-mentoring systems in healthcare environments globally. However, evidence is absent about the usability of AR technology in tele-mentoring clinical healthcare professionals in managing clinical scenarios.

### Aim

This study aims to evaluate the usability of Augmented Reality (AR) technology in tele-mentorship for managing clinical scenarios.

### Methods

This study uses a quasi-experimental design. Four experienced health professionals and a minimum of twelve novice health practitioners will be recruited for the roles of mentors and mentees, respectively. In the experiment, each mentee wearing the AR headset performs a maximum of four different clinical scenarios in a simulated learning environment. A mentor who stays in a separate room and uses a laptop will provide the mentee remote instruction and guidance following the standard protocols for the treatment proposed for each scenario. The scenarios of Acute Coronary Syndrome, Acute Myocardial Infarction, Pneumonia Severe Reaction to Antibiotics, and Hypoglycaemic Emergency are selected, and the corresponding clinical management protocols developed. Outcome measures include the mentors and mentees' perception of the AR's usability, mentorship effectiveness, and the mentees' self-confidence and skill performance.

The funders had and will not have a role in study design, data collection and analysis, decision to publish, or preparation of the manuscript.

**Competing interests:** The authors have declared that no competing interests exist.

## Ethics

The protocol was approved by the Tasmania Health and Medical Human Research Ethics Committee (Project ID: 23343). The complete pre-registration of our study can be found at https://osf.io/q8c3u/.

## Introduction

Many rural and remote areas of Australia suffer from an inadequate number of care professionals due to the maldistribution of the health workforce and the difficulties in attracting skilled health practitioners to work rurally [1–3]. The ability of the health services to support rural and remote health practitioners has been hampered by several issues, such as the high clinical loads, particularly for sole practitioners [4]; limited access to clinical supervisor [4]; difficulty accessing professional development activities or continuing education [5]; limited or no dedicated work time allocated for professional study [6]; and new graduates and sole practitioners possessing limited skills [7]. Noticeably, the national survey on non-metropolitan GPs in Australia reported that professional support was consistently ranked by rural doctors as one of their top three issues [8].

Tele-mentorship is considered to offer a solution to training and providing professional assistance at a distance. Tele-mentoring is a method in which a mentor interactively guides a mentee at a different geographic location using a technological communication device [9]. During a healthcare procedure, tele-mentorship supports an expert, remote from the treatment site, to guide a less-experienced practitioner at a different geographic location in real-time. Petridou [10] suggested that technologies can aid mentoring in rural and remote settings where geographical isolation can make communication more difficult. Advanced telecommunication technologies could enhance the effectiveness of tele-mentorships as they support the remote expert receiving all relevant information as if they were present at the remote site, and the local practitioner receiving the remote expert's instructions or guidance as if the expert was physically present.

Augmented Reality (AR) is an emerging human-computer interface and telecommunication technology. Augmented Reality is a form of immersive experience in which the real world is augmented by computer-generated three-dimensional content tied to specific locations and/or activities [11, 12]. A systematic review [13] was conducted to identify how tele-mentoring systems that incorporated AR technology have been used in healthcare environments, including its benefits and limitations. Evidence from this review supported the use of AR technology in enhancing the performance of tele-mentoring systems in healthcare environments. The benefits of AR tele-mentoring systems included a reduction in skill errors and focus shifts, improvement in task completion time and task accuracy, and positive feedback from users. The review identified weaknesses of the AR tele-mentoring systems, including not being trialled under entirely real clinical conditions or for a complete and more complex clinical scenario, and many reported only small sample sizes. Further studies that may address such weaknesses are recommended.

### Research aims and objectives

This study aims to evaluate the usability of Augmented Reality (AR) technology in tele-mentorship for managing clinical scenarios.

The study objectives are to:

- Assess mentors' and mentees' perceptions of the Augmented Reality's usability for a tele-mentorship in the management of clinical scenarios.

- Assess mentors' and mentees' scores of the effectiveness of the tele-mentorship using AR technology in the management of clinical scenarios.

- Evaluate the changes in the scores of mentees' self-confidence through the tele-mentorship using Augmented Reality technology in the management of clinical scenarios.

- Evaluate the scores of mentees' skill performance through the tele-mentorship using Augmented Reality technology in the management of clinical.

## Methods

### Study design

Due to limited research in this area, this study will test the usability of AR technology in tele-mentorship for managing clinical scenarios using a quasi-experimental design in a simulated learning environment. An overview of the study design is illustrated in Fig 1.

### Study setting

This study will be conducted at the Simulation and Clinical Education Centre, University of Tasmania (UTAS). Fig 2 illustrates the setting, which comprises a High-Fidelity (Hi-Fi) room (mentee station), a separate room (mentor station), and a control room.

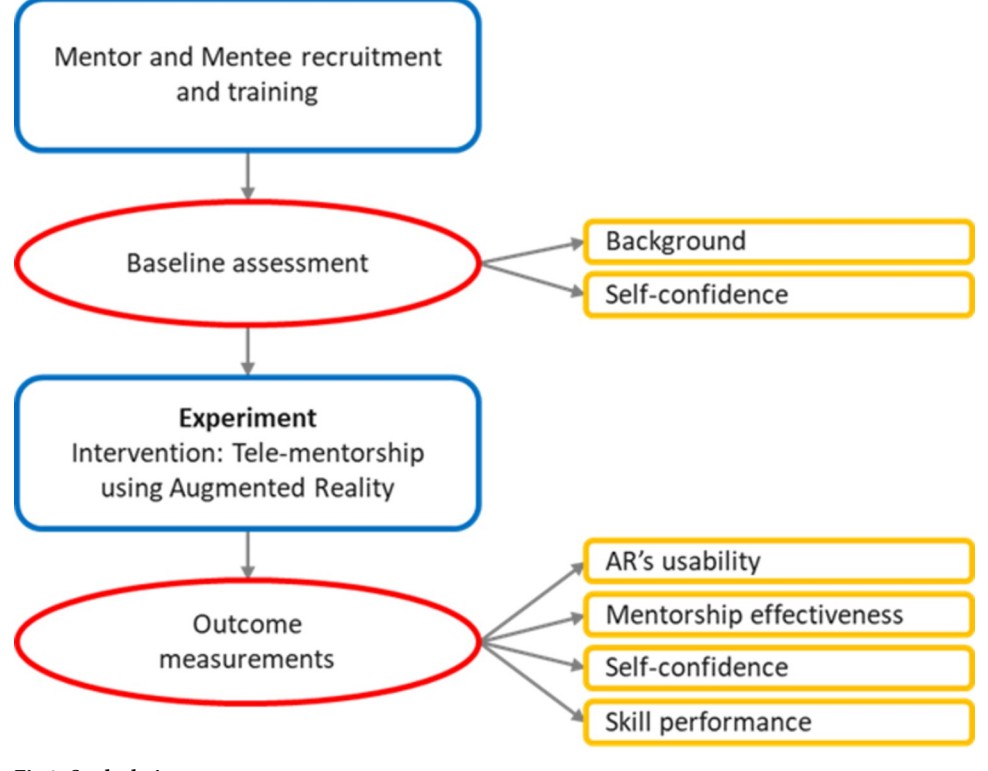

**Fig 1. Study design.**

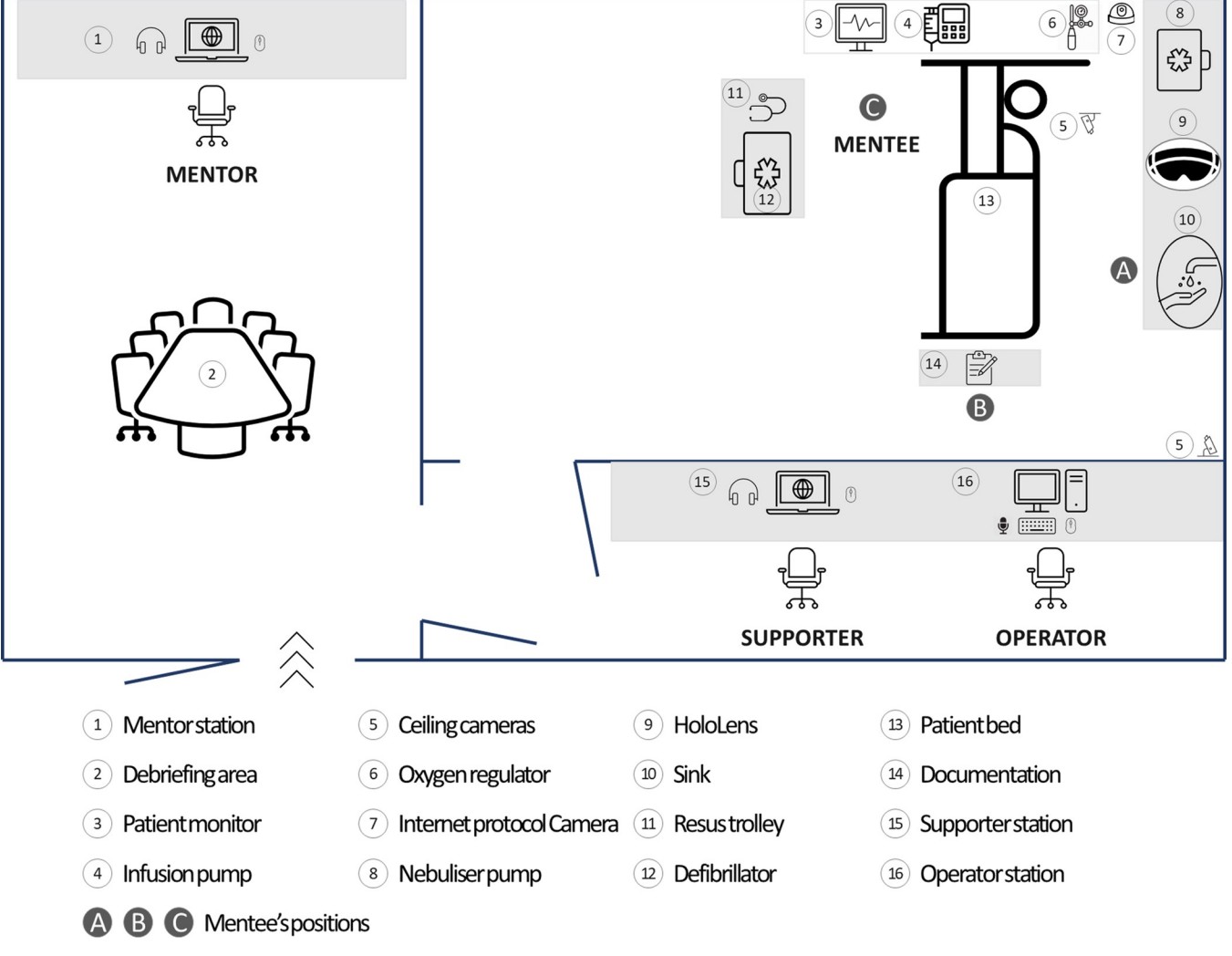

**Fig 2. Study setting.**

In the Hi-Fi room, a manikin is dressed in male clothing and lying on the bed (item 13 on Fig 2). A secure identification band with the patient's name, date of birth, and medical record number is wrapped around the manikin's wrist. Medical equipment, such as infusion pump, nebulizer pump and automated external defibrillator (AED), and medications necessary in the scenarios are also prepared. An intravenous cannula (IVC) is available to the manikin's right arm. The three leads of the electrocardiogram (ECG) are also attached to the manikin's chest and connected to the monitor mounted on the wall near the head of the patient bed. A Holo-Lens device, the newest version of an AR headset, is available ready (item 9 on Fig 2). Two cameras mounted on the ceiling and an Internet protocol camera located at the corner of the room will record all actions of the mentee during scenarios.

Adjacent to the Hi-Fi room is the mentor station. A laptop is available in the room for use by the mentor. The mentors and mentees will not see or hear each other with all connected doors closed.

In the control room, the operator controls the manikin's physiological parameters and supplies a voice dialogue of the (conscious) patient using a live microphone or pre-programmed

audio and the speakers in the Hi-Fi room. The control room is separated from the Hi-Fi room by a one-way mirror so the operator can see the mentee's actions in the Hi-Fi room.

## Participants

Two groups of participants for the study are mentors and mentees.

Eligible mentors are experienced-health professionals, such as medical doctors, registered nurses, or paramedics who are familiar with the clinical scenarios selected in this study. They may or may not have been previously involved in teaching and tutoring students in simulated learning environments. They do not necessarily need to have previous experience with AR technology, as AR devices are not widely used in health care practices.

Eligible mentees will be health practitioners or soon to be registered practitioners, such as registered nurses or paramedics who are less experienced than the mentors and less familiar with the clinical scenarios. No restrictions are established for their practical experience or previous use of AR devices. To compensate for the mentees' time to participate in the study, they will be remunerated with a gift card.

Four mentors and at least twelve mentees will be recruited to participate in the study. The mentors and mentees do not need to know each other before joining the study. Participation is voluntary. They will be asked to complete a consent form after they accept to join the study. Participants are free to withdraw from the study at any time without any consequence.

Participants have an option on whether they would like to be notified of the results of this study at enrolment to the study. If so, any publications arising from the study will be forwarded to them.

## Participant recruitment

A flyer will be placed in public media, including the university website, newspapers, and Facebook pages of the university and professional career groups to recruit participants. Interested participants will be invited to contact the student researcher, who will determine their eligibility based on the inclusion criteria. If the criteria are met, the student researcher will provide them with an information package.

The Snowball sampling method will also be used for the recruitment. Participants can recommend others to participate in the study.

## Selection of clinical scenarios

Clinical scenarios were drawn from 20 patient cases that make up the Australian and New Zealand Nursing Education scenarios which have been developed by Laerdal Australia, National League for Nursing and the Council of Deans of Nursing and Midwifery (Australia and New Zealand) [14]. The 20 scenarios have been programmed on the SimMan 3G simulators in the Hi-Fi lab, UTAS School of Nursing.

From these, four have been chosen that provided an opportunity to demonstrate unique features of the HoloLens device, such as entirely hands-free operations, real-time annotations in 3D space, and document sharing. Additional criteria for the selection of scenarios were acuity level, complexity and need for the mentee to require assistance from a remote expert to manage the scenario.

The selected scenarios (acute coronary syndrome, acute myocardial infarction, diabetic hypoglycaemic emergency, and pneumonia severe reaction to antibiotics) have been revised in accordance with the current national protocols of advanced life support [15] and Ambulance Tasmania Clinical Practice guidelines [16]. They will be programmed on the SimMan 3G simulators and then tested in the Hi-Fi lab room before use in the experiment.

## Experimental procedure

Due to the anticipated difficulties in the recruitment process and limited availability of participants due to COVID-19 restrictions, a pragmatic approach will be used in the experiment. Each mentee will be paired up with one mentor and perform all four scenarios. The multiple sessions will allow the mentor and mentee to build a mentor/mentee relationship. The sequence of scenarios will be random for each mentee. They will be asked to select one of the envelopes containing the names of the four scenarios. The selected envelop will be removed from the following selection.

Before the experiment, mentors and mentees will attend a training session individually with the same training content. The training session conducted by the first author will provide the participants with information and requisite skills regarding the study and HoloLens technology. This will include the aim and the study settings, the roles and activities of the participants, the brief of clinical scenarios, and the use of medical equipment in the Hi-Fi room. The AR technology training will brief the AR tele-mentorship setup, the use of HoloLens, and other related equipment. The mentors and mentees will familiarise themselves with the AR technology and the setup and then practice providing or receiving remote assistance via the setup.

During the experiment, each mentee will conduct primary patient observations and assessments before donning the HoloLens to connect to the mentor. The mentee will inform the mentor of the situation happening in the Hi-Fi room and as needed, request instructions from the mentor (Fig 3B). The mentor will use the laptop with a touchable screen (Fig 3A) and a video conferencing headset to communicate with the mentee in real-time, observe the situation in the Hi-Fi room and use HoloLens functionalities, such as annotation on the captured video. The mentor's instructions will adhere standard protocols for the management of each scenario. The HoloLens built-in camera, located at the front of the AR headset, will capture the real-time activities as seen by the mentee.

## Safe work procedure during COVID-19

Participants will be provided with the COVID-19 information sheet produced by UTAS College of Health and Medicine. According to the sheet, participants must be fully vaccinated with a vaccination certificate. Participants must also meet the health screening requirement

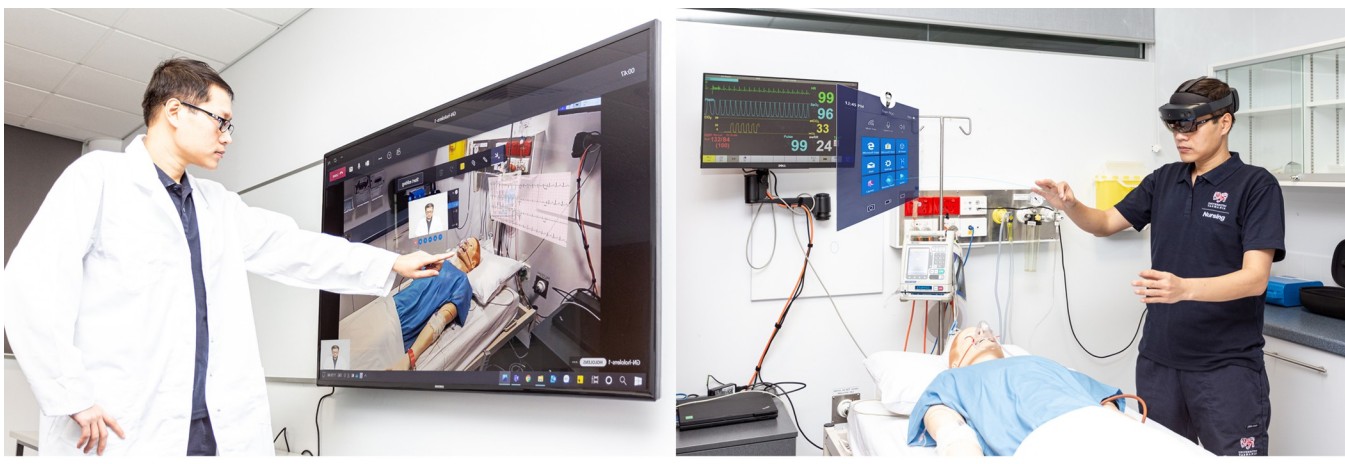

A - Mentor setup                                    B - Mentee setup

**Fig 3. Augmented reality tele-mentorship setup.** A—Mentor setup, B—Mentee setup.

with a measured temperature less than 37.5 degrees Celsius to access the Newnham campus building during the study's participation. The COVID participant screening form will be applied for all participants. They will be asked to comply with the COVID-safe plan, including the personal cleaning and physical distancing while are on campus according to the guidelines provided by the School of Health Science, UTAS.

## Outcomes

**Measures.** Different instruments will be used to evaluate the usability of the AR technology, mentorship effectiveness, self-confidence, and skill performance.

The usability of the AR setup, which is the capacity of a setup to provide a condition for its users to perform the tasks while enjoying the experience, is assessed using the scale developed by Ingrassia et al. [17]. The scale includes two parts. In the first part, the setup usability scale by Brooke et al. [18] is adopted by this study, and it is based on the ISO 9241–400 guidance on ergonomic factors (17 items). Bangor et al. [19] reported that the setup usability scale is a highly robust and versatile tool for usability professionals (Cronbach's α of 0.911). The second part of the evaluation is designed to assess setup input (15 items), setup output (7 items), fidelity (8 items), immersion (4 items), and likeability (9 items). Likert scale is used to grade each item from 1 (strongly disagree) to 5 (strongly agree). Mentors will complete 36 items because they work on a laptop instead of wearing an AR device. As mentees use the AR device, they will complete 43 items.

The effectiveness of the mentorship between mentors and mentees will be assessed in the view of mentees using the scale by Berk et al. [20] as its items seem most appropriate to evaluate responses to a wide range of mentorships as well as appropriate with the objective of the study in assessing the effectiveness of the tele-mentorship. This scale was constructed by an Ad Hoc Faculty Mentoring Committee, Johns Hopkins University School of Nursing, as an efficient, comprehensive, and standardised tool for rating the mentorship experience and, especially, the effectiveness of the mentor. The lead author of the scale granted the research team permission to use and modify the scale. The scale includes 12 items generated from the pool of positive or desirable characteristics and responsibilities of the mentor and then reviewed by the committee for their psychometric forms to determine evidence of content-related validity. The items are graded in a six-point agree-disagree continuum Likert scale from 0 (strongly disagree) to 5 (strongly agree). An item was added by the research team to get the overall perspective of the mentees. Although the mentorship effectiveness scale did not report psychometric evidence, it was cited widely and used by medical and nursing teachers [21]. This may be due to its high face and content validity or that no other psychometrically sound scale exists in medicine and the allied health field [22]. Moreover, Chen et al. [22] noticed that although some mentorship measurement tools reported its psychometric properties, the most frequently tested reliability measure was that of is internal consistency reliability. No studies included test-retest reliability or inter-rater reliability; this may imply that mentorship assessment is still in the exploration and development stage.

While the mentorship effectiveness scale developed by Berk et al. [20] is used to assess the mentees' perceptions, Dimitriadis et al. [21] modified that scale to investigate the mentors' perceptions of the mentoring relationship. This modified scale has five items. A six-point agree-disagree continuum Likert scale from 0 (strongly disagree) to 5 (strongly agree) is applied for grading these items. An item was added by the research team to get the overall perspective of the mentors. At the end of the questionnaire, three questions allow each mentor to provide personal feedback about the mentoring relationship. The research team obtained permission to use this scale granted by the lead author.

An instrument to measure self-confidence is developed with items reflecting the skills proposed for the correct treatment in each scenario. Self-confidence refers to participants (mentees)' judgments of their capabilities to achieve successful performance [23]. People with high confidence levels for a particular task are more likely to be ready and able to perform the task successfully, whereas those low in self-confidence are more likely to avoid or abandon the task. The design of the self-confidence instrument is based on similar tools used in the literature [24–26]. The self-confidence instrument requires that the participant rates his or her confidence before and after performing clinical skills in each clinical scenario. The participant uses a scale of 1 to 5 to rate his or her confidence. A score of 1 indicates no confidence at all, and a 5 indicates very high confidence.

In order to assess the mentees' skill performance, a checklist is adapted from the assessment approach of Friedman et al. [27]. The content in the checklist is taken from the proposed management of the clinical scenarios. Each item will be graded as follows grade 0 equals 'did not perform'; grade 1 equals 'inadequately performed'; and grade 2 equals 'adequately performed'.

**Data collection.** There are three points for data collection, consisting of 'before each experiment' (mentee only), 'after each experiment' (mentee only), and 'after all experiments' (mentors only). Each mentee will perform all four clinical scenarios with one assigned mentor. Each mentor will attend different experiments with different mentees. Details of the data collection process are illustrated in Table 1.

## Data analysis

During the data collection period, the research team will number each retrieved dataset and manually examine each dataset for potential concern issues. All data are then entered into the computer and analysed using SPSS software version 23.0. Descriptive statistics of frequency, percentage, mean, and standard deviation are performed to access demographic information, survey distribution, AR usability, mentorship effectiveness, self-confidence, and skill performance. The subgroups of novice/ experienced clinical practice are defined in accordance with the participants recruited to analyse the impact of tele-mentoring with respect to the participants' practical levels. The mean differences in each score are compared within a group and between groups using an independent measure t-test for parametric data, Chi-square test and Mann-Whitney U-test for non-parametric data. A two-tailed $p < 0.05$ is considered significant. Manifest content analysis is employed to analyse narrative comments in each survey.

## Data management

Hard copy data will be stored in a locked, secure location at the university for five (5) years after publication. Electronic data, including recorded videos and images, will be stored in a restricted folder accessible only by the chief investigators and the designated Archives Officer. Whilst the experiments are being conducted, the researchers will secure data, and electronic

**Table 1. Data collection.**

| Variable/measure | Who | When |
|---|---|---|
| Self confidence | Mentees | Baseline |
| Mentorship effectiveness [20] | Mentees | On completion of 4 scenarios |
| | Mentors | On completion of the experiment |
| AR usability [17] | Mentees | On completion of 4 scenarios |
| | Mentors | On completion of the experiment |
| Skill performance/checklist | Mentors | On completion of the experiment |

data will be password protected. The designated Archives Officer shall destroy the data after five (5) years. Hard copy materials will be shredded and recycled, and electronic data will be deleted from the secure servers after five years.

## Ethics approval and registration

The protocol was approved by the Tasmania Health and Medical Human Research Ethics Committee (Project ID: 23343) (See S1 File). The complete pre-registration of our study can be found at https://osf.io/q8c3u/.

## Study benefits and risks

### Benefits

The study will provide evidence on the usability and acceptability of AR technology in tele-mentorship for managing clinical scenarios. If successful and well-accepted, findings from this study may have the potentiality for applications primarily in rural and remote areas where healthcare staff may need to call upon a distant procedural expert in emergency situations. Consequently, residents living in rural and remote communities will benefit from this study. The study will also promote an understanding of using Augmented Reality technology in remote assistance between urban and suburban healthcare facilities.

Healthcare facilities that apply the recommended approach and practices derived from the findings of this study will also get benefits. On one side, metro healthcare facilities may not need to rotate as much experienced staff to rural facilities. On the other hand, rural health facilities can keep their staff on-site for specialized training instead of travelling to a metro hospital.

This study will examine critical areas in the remote assistance process lacking in the existing literature. Potentially, a new framework on tele-mentorship could be developed.

### Risks

The foreseeable risk in the studies is discomfort when participants wear the AR headset. This study will use HoloLens 2, the newest version of an AR headset [28]. According to a recently published article, this headset has been reported to be comfortable by its users [29]. However, potential discomfort is still a possibility. The headband and the adjustment wheel on the device are designed to make participants' heads fit into the device comfortably. Participants will have a chance to familiarize themselves with the headset before the experiment. Participants are to proceed with the experiments only when comfortable with the headset. Participants will be offered breaks by taking off the headset after completing each scenario and whenever requested. The contact details of the Chief Investigator and the first author will be provided if the participants need further information or assistance.

## Supporting information

**S1 File.**
(PDF)

## Acknowledgments

The authors would like to thank Christine Low, Darren Grattidge at the Centre for Rural Health, and Mark Zasadny, Annette Saunders, Kevin Wilmore, and Margaretha Yam in the

Simulation and Clinical Education Centre for preliminary discussions and expressing interest in supporting and engaging with this study.

## Author Contributions

**Conceptualization:** Dung T. Bui.

**Investigation:** Dung T. Bui.

**Methodology:** Dung T. Bui.

**Project administration:** Dung T. Bui.

**Software:** Dung T. Bui.

**Supervision:** Tony Barnett, Ha Hoang, Winyu Chinthammit.

**Validation:** Tony Barnett, Ha Hoang, Winyu Chinthammit.

**Visualization:** Dung T. Bui.

**Writing – original draft:** Dung T. Bui.

**Writing – review & editing:** Tony Barnett, Ha Hoang, Winyu Chinthammit.

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
