## [Decision Letter · Decision Letter 0]

17 Mar 2022

Usability of augmented reality technology in tele-mentorship for managing clinical scenarios - A study protocol

PONE-D-22-04739

Dear Dr. Bui,

We’re pleased to inform you that your manuscript has been judged scientifically suitable for publication and will be formally accepted for publication once it meets all outstanding technical requirements.

Kind regards,

Dylan A Mordaunt, MB ChB, MPH, MHLM, FRACP, FAIDH

Academic Editor

PLOS ONE

Additional Editor Comments (optional):

Thank you for your submission. This meets the criteria for publication.

Reviewers' comments:

Reviewer's Responses to Questions

**Comments to the Author**

1. Does the manuscript provide a valid rationale for the proposed study, with clearly identified and justified research questions?

Reviewer #1: Yes

2. Is the protocol technically sound and planned in a manner that will lead to a meaningful outcome and allow testing the stated hypotheses?

Reviewer #1: Yes

3. Is the methodology feasible and described in sufficient detail to allow the work to be replicable?

Reviewer #1: Yes

4. Have the authors described where all data underlying the findings will be made available when the study is complete?

Reviewer #1: Yes

5. Is the manuscript presented in an intelligible fashion and written in standard English?

Reviewer #1: Yes

6. Review Comments to the Author

You may also provide optional suggestions and comments to authors that they might find helpful in planning their study.

Reviewer #1: The authors propose an up-to-date technology, AR to support the professional development of health professionals in remote or rural areas. They address a relevant problem, especially in big countries with low population. The suggested method looks promising.

The description of the study protocol is detailed and sound. The authors selected validated questionnaires for data collection.

As I am not a health professional, I might be interested in more details about the clinical scenarios and the interventions to be performed by the mentees. If the other reviewers or the editor ask for a revision, you might also consider adding some examples. Otherwise, the paper can be published as it is.

Note: a Word comment is left on page 14.

7. PLOS authors have the option to publish the peer review history of their article (what does this mean?). If published, this will include your full peer review and any attached files.

Reviewer #1: No

---

## [Editor Report · Acceptance letter]

21 Mar 2022

PONE-D-22-04739 

Usability of augmented reality technology in tele-mentorship for managing clinical scenarios - A study protocol 

Dear Dr. Bui:

I'm pleased to inform you that your manuscript has been deemed suitable for publication in PLOS ONE. Congratulations! Your manuscript is now with our production department. 

Kind regards, 

on behalf of

Dr. Dylan A Mordaunt 

Academic Editor

PLOS ONE